## [Peer Review File · Communications Biology]

Reviewers' comments:

Reviewer #1 (Remarks to the Author):

In this revised manuscript and response, Chapin and colleagues address the points raised regarding the earlier version of their manuscript, in particular my primary concern with the ABBS method regarding the accuracy of quantitation of methylation. However, I feel that the similarity in quantitation of methylation is still overstated in both the manuscript and the response, the differences are not sufficiently noted or discussed, and the tendency of ABBS to report higher methylation levels should be clearly stated in the manuscript.

As suggested, the authors now provide an additional analysis of the correlation between WGBS- and ABBS-based quantitation of methylation level at individual CpG sites that had between ~20-80% methylation in either WGBS or ABBS (Supp. Fig. 2d). As anticipated, this shows significantly lower correlation between the two methods (PCC = 0.82) compared to when the many fully methylated or unmethylated CpG sites are also considered (PCC = 0.99, Fig. 2a). To address potential influence of restricted range effect, the authors consider equal numbers of CpGs of differing WGBS-based methylation quantitation, which shows a high correlation (PCC = 0.93). However, this analysis does not fully address how well ABBS estimates intermediate levels of methylation, as this reported value will still be influenced by inclusion of very lowly methylated and very highly methylated CpGs (truly completely methylated and completely unmethylated CpGs will only ever be detected as such, excluding failures in bisulfite conversion or sequencing errors). The lower correlation in methylation level estimates between ABBS and WGBS at sites of intermediate methylation should be noted in the manuscript text. Furthermore, Supp. Fig. 2d (both panels) clearly shows that ABBS tends to report a higher level of methylation compared to WGBS. This tendency should also be mentioned in the main text of the manuscript.

In both the response and the revision, the authors make repeated short statements about the agreement between ABBS and WGBS in the quantitation of methylation, but there is little depth or discussion around these points. For example: "Importantly, there was an excellent agreement in the methylation levels measured by ABBS and WGBS (Fig. 2a, Sup. Fig. 2d), establishing ABBS as an accurate method to quantify cytosine methylation genome-wide.". Or in their response: "ABBS in fact does allow methylation to be accurately determined for each covered site. In fact, the % methylation measurements made by ABBS are nearly identical to those obtained with WGBS".

Figure 2d also appears to show that the quantitation of intermediate methylation by ABBS is different to WGBS, as ABBS seems to show higher methylation values in general within these regions. This area seems to have very low levels of methylation in general (0-15%). I feel that there is a discordance between aspects of the data presented like this, and statements in the manuscript and response like "ABBS is able to accurately measure low methylation levels at many C sites (e.g. Fig. 2d)".

Supp. Fig. 2e: methylation sites and estimated levels are shown for regions of ~300 kb and 1 MB, in which the ABBS and WGBS methylation patterns match somewhat, on a regional level, but it would appear that they are quite different at the single site level, as the patterns look substantially different. At this magnification of the browser it is hard to know whether this is due to differences in coverage, or whether they reflect inherent differences in the ability of the two methods to detect sites of methylation. A greater magnification to see correspondence at individual bases in a much smaller region would be helpful. Furthermore, it is hard to judge how well this reflects the concordance between the methods more generally. Availability of the datasets in a genome browser would help assessment of this.

As described above, the data presented appears to show that there are some differences in quantitation between ABBS and WGBS, but these are not accurately communicated in the manuscript. Every method has some biases, that is not unexpected, but what is important is to clearly communicate them when reporting a method, so that potential users are aware.

Reviewer #2 (Remarks to the Author):

I am satisfied with the revisions.

Carlsbad, 03/29/2022.

Response to Referees

We would like to thank the Referees for the insightful and constructive comments, as the manuscript is considered for publication in *Communications Biology*.

As detailed below, we addressed the Reviewers' comments and questions and believe that this version of the manuscript provides a fair assessment of the ABBS method, including notes on potential weaknesses of the method.

Specific responses are below. As a matter of reference, we added numbers (shown in blue text) to some of the Referee's comments.

Reviewer #1 (Remarks to the Author):

1. In this revised manuscript and response, Chapin and colleagues address the points raised regarding the earlier version of their manuscript, in particular my primary concern with the ABBS method regarding the accuracy of quantitation of methylation. However, I feel that the similarity in quantitation of methylation is still overstated in both the manuscript and the response, the differences are not sufficiently noted or discussed, and the tendency of ABBS to report higher methylation levels should be clearly stated in the manuscript.

As suggested, the authors now provide an additional analysis of the correlation between WGBS- and ABBS-based quantitation of methylation level at individual CpG sites that had between ~20-80% methylation in either WGBS or ABBS (Supp. Fig. 2d). As anticipated, this shows significantly lower correlation between the two methods (PCC = 0.82) compared to when the many fully methylated or unmethylated CpG sites are also considered (PCC = 0.99, Fig. 2a). To address potential influence of restricted range effect, the authors consider equal numbers of CpGs of differing WGBS-based methylation quantitation, which shows a high correlation (PCC = 0.93). However, this analysis does not fully address how well ABBS estimates intermediate levels of methylation, as this reported value will still be influenced by inclusion of very lowly methylated and very highly methylated CpGs (truly completely methylated and completely unmethylated CpGs will only ever be detected as such, excluding failures in bisulfite conversion or sequencing errors). The lower correlation in methylation level estimates between ABBS and WGBS at sites of intermediate methylation should be noted in the manuscript text. Furthermore, Supp. Fig. 2d (both panels) clearly shows that ABBS tends to report a higher level of methylation compared to WGBS. This tendency should also be mentioned in the main text of the manuscript.

In both the response and the revision, the authors make repeated short statements about the agreement between ABBS and WGBS in the quantitation of methylation, but there is little depth or discussion around these points. For example: "Importantly, there was an excellent agreement in the methylation levels measured by ABBS and WGBS (Fig. 2a, Sup. Fig. 2d), establishing ABBS as an accurate method to quantify cytosine methylation genome-wide.". Or in their response: "ABBS in fact does allow methylation to be accurately determined for each covered site. In fact, the % methylation measurements made by ABBS are nearly identical to those obtained with WGBS".

We agree that ABBS moderately overestimates intermediate methylation levels at several CpG sites, and that this should clearly be noted in the manuscript. We have now revised the main text related to Fig. 2a and supp Fig. 2d as follows:

(1) We softened the statement “there was an excellent agreement in the methylation levels measured by ABBS and WGBS” by replacing “excellent” with “strong” (lines 124-125).

(2) We removed the sentence “(...) ABBS as an accurate method to quantify cytosine methylation genome-wide” (previously, lines 142-143).

(3) We described the higher level of methylation observed in ABBS for certain CpG as well as the lower correlation between ABBS and WGBS for methylation levels between 20-80% and the restricted range effect (lines 127-140).

The new paragraph now reads:

“Importantly, there was a strong agreement in the methylation levels measured by ABBS and WGBS ($PCC=0.99$, Fig. 2a). We also note that ABBS produces slight overestimates relative to WGBS at several cytosines (Sup. Fig. 2d, see points above the diagonal). Notably, the bi-modal distribution of methylation in the human genome results in abundant extreme values (very low and very high methylation levels), which can skew the correlation values. Therefore, we assessed whether ABBS produces accurate measurements at sites of intermediate methylation levels (i.e. between 20% and 80% methylation). At these sites, we found a good but decreased correlation values ($PCC=0.82$, Sup. Fig. 2d), which may due in part to the effect of restricted range on correlation. To circumvent this issue, we also re-calculated the Pearson correlation coefficient in a subset where cytosines are homogeneously distributed on the X axis, by sampling equal amounts (5,000) of cytosines in each of the twenty 5%-wide bins spanning the full range of methylation (i.e. 5000 cytosines in the 0-5% range, 5000 cytosines in the 5-10% range, ..., 5000 cytosines in the 95-100% range). In this homogeneously distributed dataset, the methylation levels measured by ABBS strongly correlated with those measured by WGBS ($PCC = 0.93$, including hypo- and hypermethylated sites, Supp. Fig. 2d, right panel).”

2. Figure 2d also appears to show that the quantitation of intermediate methylation by ABBS is different to WGBS, as ABBS seems to show higher methylation values in general within these regions. This area seems to have very low levels of methylation in general (0-15%). I feel that there is a discordance between aspects of the data presented like this, and statements in the manuscript and response like "ABBS is able to accurately measure low methylation levels at many C sites (e.g. Fig. 2d)".

We thank this Referee for noting this. We have added on lines 155-156: “(...) although ABBS seems to show slightly higher methylation values than WGBS in general (Fig. 2d).”

3. Supp. Fig. 2e: methylation sites and estimated levels are shown for regions of ~300 kb and 1 MB, in which the ABBS and WGBS methylation patterns match somewhat, on a regional level, but it would appear that they are quite different at the single site level, as the patterns look substantially different. At this magnification of the browser it is hard to know whether this is due to differences in coverage, or whether they reflect inherent differences in the ability of the two methods to detect sites of methylation. A greater magnification to see correspondence at individual bases in a much smaller region would be helpful. Furthermore, it is hard to judge how well this reflects the concordance between the methods more generally. Availability of the

datasets in a genome browser would help assessment of this.

We have now added 3 magnification scales down to single base resolution in supp. Fig. 2e. We also agree with this referee's above comment in that genome browser regions were previously showing sites that are lowly methylated (0-15% methylation) and have thus selected other regions where the full range of methylation levels (0-100%) can be observed in Fig. 2d and supp. Fig. 2e. In addition, we have tried to upload the data to the UCSC genome browser but were unfortunately unsuccessful due to file size. We apologize for the inconvenience and hope that this reviewer finds the new Fig. 2d and supp. Fig. 2e. sufficient for publication.

As described above, the data presented appears to show that there are some differences in quantitation between ABBS and WGBS, but these are not is accurately communicated in the manuscript. Every method has some biases, that is not unexpected, but what is important is to clearly communicate them when reporting a method, so that potential users are aware.

Reviewer #2 (Remarks to the Author):

I am satisfied with the revisions.

We thank Referee #2 very much for re-reviewing this manuscript.